# Vector Representations of Vessel Trees

**James Batten**[1,2]                                    J.BATTEN@IMPERIAL.AC.UK
**Michiel Schaap**[1,2]                                MSCHAAP@HEARTFLOW.COM
**Matthew Sinclair**[1,2]                            MSINCLAIR@HEARTFLOW.COM
**Ying Bai**[2]                                                  YBAI@HEARTFLOW.COM
**Ben Glocker**[1,2]                                  B.GLOCKER@IMPERIAL.AC.UK

[1] *Imperial College London, Exhibition Road, London*

[2] *HeartFlow, 331 E Evelyn Ave, Mountain View, California*

**Editors:** Under Review for MIDL 2025

## Abstract

We introduce a novel framework for learning vector representations of tree-structured geometric data focusing on 3D vascular networks. Our approach employs two sequentially trained Transformer-based autoencoders. In the first stage, the Vessel Autoencoder captures continuous geometric details of individual vessel segments by learning embeddings from sampled points along each curve. In the second stage, the Vessel Tree Autoencoder encodes the topology of the vascular network as a single vector representation, leveraging the segment-level embeddings from the first model. A recursive decoding process ensures that the reconstructed topology is a valid tree structure. Compared to 3D convolutional models, this proposed approach substantially lowers GPU memory requirements, facilitating large-scale training. Experimental results on a 2D synthetic tree dataset and a 3D coronary artery dataset demonstrate superior reconstruction fidelity, accurate topology preservation, and realistic interpolations in latent space. Our scalable framework, named VeTTA, offers precise, flexible, and topologically consistent modeling of anatomical tree structures in medical imaging.

**Keywords:** Tree Autoencoders, Recursive Decoding, Coronary Artery Modeling

## 1. Introduction

Vector embeddings are powerful representations for mapping data from various modalities into a universal format. Although autoencoders have been widely used to learn embeddings from images, text, and audio, the problem of encoding and decoding graph-structured geometry remains challenging (Zhu et al., 2022). For example, representing 3D anatomical structures such as coronary arteries is particularly difficult due to their highly variable topology.

Learning to encode precise topological information into a single vector representation in a variational setting requires a particularly powerful and expressive encoder. Transformers (Vaswani, 2017) are well-suited to this task because they effectively leverage large-scale training compute and extensive datasets. Furthermore, the architecture can efficiently model long-range dependencies to better capture fine-grained geometric details. These capabilities enable transformer-based encoders to model the full complexity of branching structures in a global latent code, preserving the geometry of tree-structured data.

Vector embeddings provide compact representations that facilitate efficient integration across modalities and downstream tasks. In medical imaging, learning these embeddings

can enable more accurate predictions of vessel geometry directly from images, improving patient-specific analyses like hemodynamic assessments or coronary artery disease diagnosis. Beyond reconstruction tasks, generative modeling approaches have gained traction in medical imaging for data augmentation and shape analysis. By capturing the distribution of anatomical variations in a latent space, generative models can create realistic synthetic shapes to expand training data or explore morphological changes across a patient population (Rasal et al., 2023). These avenues highlight the importance of learning powerful, flexible encodings for complex anatomical structures.

In this paper, we demonstrate that an autoencoder capable of decoding continuous geometric representations of trees can accurately reconstruct samples not seen during training. This approach is valuable in scenarios where models are first pre-trained on large geometric datasets and then applied to downstream tasks requiring high precision (e.g., image-to-geometry problems), where memory constraints result in significant limits to scalability. Our method, named VeTTA (Vessel Tree Transformer Autoencoder), employs a recursive decoder, which provides a topological guarantee that the output geometry is tree-structured. The latent code describes a recursive "program" for decoding the branching structure across multiple forward passes through the model, enabling the reconstruction process to effectively leverage the capabilities of the decoder at inference-time. Finally, we show that when interpolating latent vectors between previously unseen samples, our method generates plausible and topologically-correct trees, indicating that our model learns embeddings in a semantically meaningful latent space.

## 2. Related Work

Generative modeling of 3D geometry has gained momentum in numerous applications, such as molecule design (Shi et al., 2020), vessel graph synthesis (Schneider et al., 2012) and shape representation (Lemeunier et al., 2022). Some generative methods involve volumetric (Brock et al., 2016) or mesh-based operations (Liu et al., 2023; Siddiqui et al., 2023), while more recent approaches exploit implicit functions to capture geometric detail (Genova et al., 2019). Transformers have emerged as powerful sequence-to-sequence models for 3D data, where positional embeddings or local structural features can be encoded directly (Mialon et al., 2021).

Graph-centric generative models typically aim to produce discrete connectivity along with position information, prominent in vascular network generation (Prabhakar et al., 2024; Feldman et al., 2023). They often adopt variational inference (Gómez-Bombarelli et al., 2018) or adversarial training (De Cao and Kipf, 2018) to learn expressive latent representations, supporting tasks like structure optimization (Simonovsky and Komodakis, 2018) or hierarchical composition (Jin et al., 2018).

Another line of work, relevant to representation learning of curvilinear structures, focuses on trajectory and sequence modeling with autoencoders, including motion analysis (Jadhav and Farimani, 2021), agent trajectory prediction (Chen et al., 2023), or large-scale spatio-temporal synthesis (Chen et al., 2021). Approaches like hierarchical Reinforcement Learning and trajectory-level variational autoencoders have also been explored to learn stable latent spaces for planning (Co-Reyes et al., 2018), while specialized architectures can discover compact embeddings for trajectory clustering (Olive et al., 2020). In parallel, shape-specific

generative models—especially for blood vessels (Wolterink et al., 2018)—highlight the need for topologically consistent representations, whether through tube-like primitives, implicit constraints, or recursion on branch structures.

## 3. Method

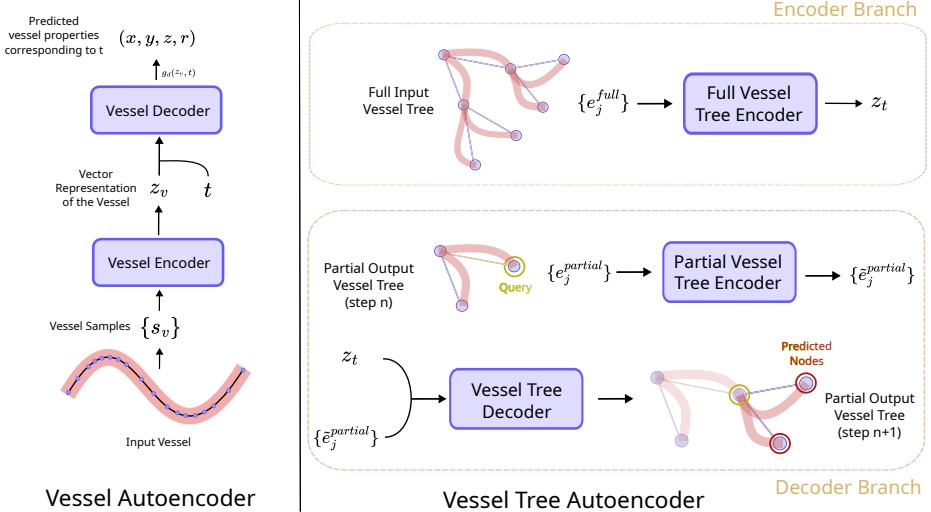

Figure 1: Proposed method diagram. First stage (left): the Vessel Autoencoder which encodes a vessel segment into an embedding $z_v$. Second stage (right): the Vessel Tree Autoencoder which encodes a whole tree into a vector representation $z_t$.

In our framework, input vessel segments are represented by sequences of n points $(x, y, z, r, t)$, where $(x, y, z)$ denote spatial coordinates, $r$ is the local radius, and $t$ is the normalized position along the segment. Before encoding, we lift these coordinates into sinusoidal Fourier features (Tancik et al., 2020; Sitzmann et al., 2020; Benbarka et al., 2022), capturing high-frequency geometric variations and improving reconstruction accuracy.

Our Vessel Tree Transformer Autoencoder (VeTTA) is evaluated in 2D and 3D settings. In 2D, vessel segments are represented as directed edges between discrete nodes without explicit curvature or radius. In 3D (VesselTrees dataset), full curvilinear centerlines and radii are encoded via a dedicated Vessel Autoencoder. We test both standard and variational autoencoder variants in both settings. Our representation models vessels as generalized cylinders parameterized by coordinates and radius, sufficient for accurate mesh reconstruction. Throughout this paper, "topology" explicitly refers to discrete connectivity between nodes and edges (e.g., branching patterns).

**Vessel Autoencoder.** We begin by training a *Vessel Autoencoder* to represent a single generalized cylinder of shape $(n)$, where each sample $(x, y, z, r, t)$ is lifted to sinusoidal Fourier features (Tancik et al., 2020) before being processed by a transformer encoder and small MLP layers. Let $z_v \in \mathbb{R}^{64}$ be the final encoded vessel embedding. A 2-layer MLP

$f_d(\cdot)$ then predicts a residual to a linear interpolation between the first and second node positions (and radii), as

$$g_d(z_v, t) \;=\; a \;+\; (b-a) \cdot t \;+\; f_d(z_v, t) \cdot m(t), \tag{1}$$

where $t \in [0,1]$ and $(a,b)$ are the normalized endpoints with the radius terms: $((p_a, r_a), (p_b, r_b))$. During training, we set $m(t) = 1$ for stable optimization. During evaluation, we use $m(t) = 0.5^{-0.2} \cdot t^{0.1}(1-t)^{0.1}$, which ensures exact matching at vessel endpoints $(m(0) = m(1) = 0)$, enabling precise endpoint alignment and smooth curvature reconstruction.

We normalize each vessel by placing $p_a$ at the origin, $p_b$ on the unit sphere and linearly rescaling the radius such that $r_a = 1$. We split each vessel into 64 segments of equal *Gaussian-smoothed curvature* and sample one point per segment for the encoder. We further re-weight the reconstruction loss to emphasize longer and more tortuous vessels (see Appendix for details). For each batch sample, scalar values $t$ are then drawn to compare the reconstruction $g_d(z_v, t)$ with the ground-truth $(x, y, z, r)$ via

$$\mathcal{L}_{\text{Vessel}} \;=\; \big\| g_d(z_v + \epsilon, t) \;-\; (x, y, z, r) \big\|_2^2, \tag{2}$$

where a small amount of Gaussian noise $\epsilon$ of standard deviation 0.05 is added to $z_v$ to locally regularize the latent space. Because $f_d(\cdot)$ is a small MLP with a GeLU activation, it yields a continuous path ideally suited for vascular modeling.

**Vessel Tree Autoencoder.** In the second stage, we train a *Vessel Tree Autoencoder* on a *tree connectivity structure*, i.e. a set of directed edges. Each edge comprises node coordinates (again lifted to Fourier features), topological attributes, and in the 3D setting a concatenated $z_v$ from the Vessel Autoencoder. We pass the full set of edges $\{e_i^{\text{full}}\}$ through a transformer to produce a global tree representation $z_t = \text{EncoderBranch}(\{e_i^{\text{full}}\})$ which can be viewed as a "program" describing how to reconstruct the tree (c.f. Fig 1).

In the *variational* configuration, instead of predicting a single vector $z_t$, the encoder outputs two vectors $(z_{\mu,t}, z_{\sigma,t})$, and we sample $z_t$ via the reparametrization trick. We then add a Kullback-Leibler divergence term to the total loss:

$$\mathcal{L}_{\text{KL}} \;=\; \lambda_{\text{KL}} \, D_{\text{KL}}\big(\mathcal{N}(z_{\mu,t}, z_{\sigma,t}^2) \,\|\, \mathcal{N}(\mathbf{0}, \mathbf{I})\big), \tag{3}$$

where $\lambda_{\text{KL}}$ is a hyperparameter. This term is present only in the variational setting.

**Recursive Decoding.** During inference, we decode the tree in a recursive fashion. At each step, we choose one node in a *partial* tree whose children remain unknown, marking it with a binary "query" flag in the directed edges $\{e_j^{\text{partial}}\}$. A transformer encoder processes these partial edges, concatenating $z_t$ to each edge embedding as context. To handle the special case where the partial tree is empty at the start of decoding (i.e. predicting the root node), we add a learned "start token" to avoid an entirely empty input. After the root node is predicted in the first step, we use a "semi-edge" that connects the root node to itself so that the partial tree still has at least one valid edge for the encoder to process (see Appendix for details). Once all the nodes in the partial tree have been expanded, the entire tree is decoded and recursive process halts.

The encoded partial tree (plus $z_t$) serves as the "memory" of a transformer decoder whose input is a fixed set of *slots* (learned vectors). The decoder uses cross-attention

to extract relevant information from $z_t$ and $\{e_j^{\text{partial}}\}$, and produces predictions for each slot: $\{(s_p, s_t)\} = \text{DecoderBranch}\big(\{e_j^{\text{partial}}\}, z_t\big)$. Here $s_p$ and $s_t$ denote, respectively, the predicted positional and topological attributes; in 3D, a vessel embedding, a log radius and a skip-vessel flag are also predicted for each slot. Because any node in the tree can have at most two children, the network learns to bind its set of predicted slots to either one or two ground-truth child nodes. We then *cluster* the slot predictions to form discrete child nodes (the cluster count is given by the query node's predicted topology). This clustering approach is described in more detail in the Appendix. In the experiments with our proposed model, we use 32 slots.

To reconstruct the continuous vessel geometry from each predicted child, we interpolate between the parent $(p_a, r_a)$ and child $(p_b, r_b)$ and refine curvature via Eq. 1. In practice, we reweight different properties in the decoder loss (position, topology, radius, and so on) to balance the relative scales of each quantity. We introduce a binary *skip-vessel flag* to handle short vessel segments ($\leq 2.5$mm). Edges flagged as skip-vessels exclude vessel embeddings during training, and at inference, these segments are reconstructed by linear interpolation between their endpoints.

**Recursive Reconstruction Loss.** To train this decoder, we define $\Phi(\cdot)$ as a lifting function that maps positional coordinates, topological vectors, and (for 3D) vessel embeddings into a high-dimensional space, enabling a distance measure between predicted slot-vectors $S = \{s_i\}$ in the lifted domain and target nodes $G = \{g_j\}$ in the original domain. We form the cost matrix $\|s_i - \Phi(g_j)\|_2^2$ and compute a custom matching in both directions to produce the matrices $L$ and $R$, such that each slot is matched to exactly one target and each target has as least $k$ slots matched to it (In our experiments we set $k$ to 3, see Appendix for more details). By minimising the reconstruction loss, the model learns to recursively decode valid tree-structured reconstructions from a single global embedding. The total reconstruction loss for a single decoding step is:

$$\mathcal{L}_{\text{Tree}} = \frac{1}{|L|} \sum_{s_i \in S}^{s} \sum_{g_j \in G}^{t} L_{ij} \, \|s_i - \Phi(g_j)\|_2^2 \;+\; \frac{1}{|R|} \sum_{s_i \in S}^{s} \sum_{g_j \in G}^{t} R_{ij} \, \|s_i - \Phi(g_j)\|_2^2, \qquad (4)$$

**Variational Autoencoder.** In the variational setting, we add $\lambda_{\text{KL}} \, D_{\text{KL}}(\cdot)$ as described above to obtain the full training objective. In the variational case, we can also linearly interpolate between two distinct $z_t$ samples to synthesize novel plausible morphologies.

## 4. Experimental Setup

We train autoencoder models on two datasets. The first is the publicly available SSA dataset (https://zenodo.org/record/10076802), which contains 2D synthetic trees (15248 training trees, 3812 test trees) with associated segmentations. The second is the VesselTrees dataset, composed of semi-automatically annotated 3D coronary meshes (and centerlines) derived from cardiac CT angiography (CTA) through HeartFlow's commercial processing pipeline. It includes vessel radii at discrete nodes along the centerline tree. In both datasets, we limit each node in the tree connectivity structure to having either 0, 1, or 2 children. The VesselTrees dataset is divided into 3996, 754 and 250 full coronary trees for the train, test and val splits respectively.

For both the SSA and VesselTrees datasets, we conduct two sets of experiments in which each model is trained as a standard autoencoder and as a variational autoencoder (respectively indicated with suffixes -AE and -VAE). On the VesselTrees dataset, we subsample 10 random "sub-trees" per full coronary tree, with a maximum arc length from root to leaf of 60mm. We exclude both subsampled vessel trees that have fewer than 3 segments and samples with no bifurcations. In order to train the Vessel Autoencoder, we crop single vessels in the range 2.5 to 40mm. The subsampled vessel trees contain segments between 0 and 40mm in length. Segments smaller than 2.5mm in length are explicitly represented by a skip-vessel flag.

On the 2D SSA dataset, we compare the proposed autoencoder model, VeTTA, against a convolutional autoencoder baseline (Conv-2D). We use two variants of this convolutional architecture: one with batch normalization (Ioffe, 2015) and ReLU activations, and another with GroupNorm (Wu and He, 2018) and GELU (Hendrycks and Gimpel, 2016). Both baselines are trained to encode and decode the 2D segmentation masks. After decoding, we take the largest connected component of the output and skeletonize it to extract the predicted centerline. By contrast, our method directly predicts the tree structure via recursive decoding; we then sample 100 points down each predicted edge to form a centerline. Centerline-based metrics are computed to compare the predicted trees to the ground truth. All models trained on the SSA dataset use a latent code size of 256.

On the 3D VesselTrees dataset, we compare VeTTA against three convolutional architectures: GDVM-AE (Brock et al., 2016), Conv-3D-AE (our extension of GDVM-AE with residual connections and GroupNorm), and VesselVAE (Feldman et al., 2023). We train each model to learn shape representations from $128^3$ voxel grids. For GDVM-AE and Voxel-AE, we convert the resulting 3D volumes into meshes with marching cubes, and then extract the centerline through skeletonization. We also generate a point cloud from the segmentation by sampling points within the volume. In contrast, our method first predicts continuous centerline curves (with radius) structured in a tree. To obtain a reconstructed mesh, we create a surface point cloud from these curves by placing points around each predicted radius; we then apply Poisson surface reconstruction (Kazhdan et al., 2006) to produce a watertight mesh. From this mesh, we additionally derive 3D volumes and volumetric point clouds (by uniform sampling in the interior). These four representations—mesh, centerline, point cloud, and voxel grid—are evaluated for both the proposed approach and the convolutional baselines (except for VesselVAE, where we do not compute the dice score since the output mesh is not always watertight). All models trained on the VesselTrees dataset use a latent code of size 8192.

We train the Vessel Autoencoder for 140k steps with a batch size of 1750. The learning rate is linearly increased to $5 \times 10^{-5}$ and then exponentially decayed by a factor of 10 every 50k steps. The Vessel Tree Autoencoder is trained for 250k steps with a batch size of 1100, linearly ramping up the learning rate to $10^{-5}$ with a factor 10 decay every 200k steps. We use the AdamW optimizer (Loshchilov, 2017) to train all the models described in this paper. For the variational models, we weight the KL-divergence term by a factor $\lambda_{KL} = 1\mathrm{e}{-6}$. All models are implemented using Pytorch (Paszke et al., 2019) and are trained on a single NVIDIA V100 32GB GPU. Further architectural and implementation details for both our approach and the baselines can be found in the Appendix. The code to train these models will be made publicly available.

## 5. Results and Discussion

Compared to mesh or volumetric segmentation representations, our compact vector embeddings efficiently summarize complex and continuous geometric structures into fixed-length formats, facilitating integration into downstream analysis without discretization artifacts. Our approach directly leverages explicitly provided curvilinear geometry, highlighting the advantages of mapping centerline structures to and from vector embeddings.

| Model | CHD $\downarrow$ ($\times 10^1$) | ACD $\downarrow$ ($\times 10^2$) | CF1 $\uparrow$ |
|---|---|---|---|
| Conv-2D-AE (BN) | 0.327 | 0.522 | 0.417 |
| Conv-2D-AE (GN) | 0.644 | 0.707 | 0.328 |
| **VeTTA-AE** | **0.184** | **0.348** | **0.504** |
| Conv-2D-VAE (BN) | 4.09 | 6.94 | 0.0848 |
| Conv-2D-VAE (GN) | 0.241 | 0.493 | 0.415 |
| **VeTTA-VAE** | **0.225** | **0.405** | **0.439** |

Table 1: Performance comparison of different models on the SSA Dataset, with the metrics CHD (Centerline Hausdorff distance), ACD (Avg. centerline distance), and CF1 (Centerline F1 score, using a distance threshold of 0.05)

**Comparison on SSA Dataset.**

As summarized in Table 1, our proposed method produces reconstructions that outperform the baseline convolutional autoencoders on the synthetic 2D tree dataset, in both the standard and variational configurations. All methods here use the same latent size of 256. The aim of the 2D experiment is to demonstrate that our approach can accurately handle tree-structured data in the simplest possible setting, and both our model and the convolutional baselines produce highly accurate reconstructions. The 2D trees are contained within the $[0, 1] \times [0, 1]$ domain; all distance metrics are computed in this space.

We observe a contrast in performance between the two convolutional baselines, one using BatchNorm with ReLU (Conv-AE (BN)) and the other using GroupNorm+GELU (Conv-AE (GN)). In the classic autoencoder setting, Conv-AE (BN) exhibits slightly better metrics. However, in the variational setting, the BatchNorm model fails to reconstruct reliably, whereas Conv-AE (GN) is on par with our proposed method. Motivated by this discrepancy, we use the GN variant for our 3D volumetric baselines so that performance remains stable across both configurations.

**Comparison on VesselTrees Dataset.**

In the 3D setting (Table 2), all models share a larger bottleneck dimension of 8192. To handle the variability in vessel tree size and location, each sample is normalized during training and inference by computing the bounding box and applying the similarity mapping its longest side to the range $[-0.25, 0.25]$ and its center to the origin. During evaluation, this normalization mapping is inverted to measure final reconstruction errors in millimeters. Our method demonstrates higher Dice overlap, improved Hausdorff distances, and superior centerline-based metrics compared to the convolutional baselines. Visual examples (see Figure 2) illustrate that VeTTA recovers topologically correct, plausible 3D

| Model | Dice ↑ | Mesh HD ↓ | ASD ↓ | ACD ↓ | APCD ↓ |
|---|---|---|---|---|---|
| GDVM-AE | 46.5 | 5.42 | 1.10 | 1.34 | 0.946 |
| Conv-3D-AE | 71.8 | 2.22 | 0.525 | 0.587 | 0.527 |
| **VeTTA-AE** | **85.4** | **2.09** | **0.288** | **0.326** | **0.478** |
| VesselVAE | - | 15.9 | 7.13 | 2.70 | 2.47 |
| GDVM-VAE | 33.0 | 8.97 | 1.89 | 1.64 | 1.60 |
| Conv-3D-VAE | 51.3 | 2.87 | 0.830 | 0.776 | 0.650 |
| **VeTTA-VAE** | **78.5** | **2.50** | **0.377** | **0.458** | **0.537** |

Table 2: Performance comparison of different models on the VesselTrees Dataset, with the metrics Dice, Mesh HD (Mesh Hausdorff distance), ASD (Avg. Surface Distance), ACD (Avg. Centerline Distance) and APCD (Avg. Point Cloud Distance).

coronary trees, and outputs continuous centerlines. By contrast, the 3D CNN baselines rely on skeletonization of their volumetric predictions to compute the centerlines, and these segmentations often suffer small disconnects or spurious branches on the voxel grid, which degrades reconstruction accuracy. Qualitatively, we observed that vessel reconstructions obtained using our proposed VeTTA method generally appeared more realistic, exhibiting smoother vessel surfaces and more consistent continuity compared to reconstructions from convolutional baselines (GDVM-VAE and Conv-3D-VAE). The convolution-based methods often produced reconstructions with voxel discretization artifacts, resulting in rougher surfaces and occasional disconnected vessel segments. In contrast, our vector-based approach consistently generated smoother, more anatomically plausible vessel geometries. We observed larger relative improvements in Dice scores compared to Hausdorff distances (HD). This discrepancy arises because Dice scores aggregate overall volumetric overlap, whereas HD is highly sensitive to isolated errors, thus occasionally highlighting outliers.

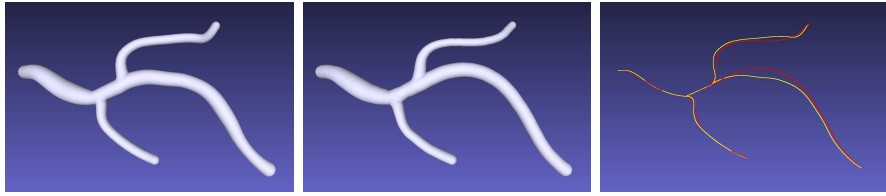

Figure 2: Example reconstruction with the proposed model. Left: ground truth mesh from the VesselTrees dataset. Middle: proposed reconstruction. Right: Overlay of the ground truth centerline (yellow) and the reconstructed centerline predicted by the proposed model (red).

Although we also compare to VesselVAE, its primary purpose is generative modeling of vascular networks rather than precise reconstruction. In contrast to the parallel transformer-based encoder in our proposed model (which encodes the structure in a single forward pass),

VesselVAE employs a multi-step encoder which may pose challenges for backpropagation, potentially explaining its weaker reconstruction performance in our experiments.

**Latent Interpolations.** Beyond quantitative metrics, both our 2D and 3D models produce smooth, semantically meaningful interpolations between previously unseen trees, as shown in Figure 3. The recursive decoding architecture ensures topologically valid intermediate shapes when linearly mixing latent codes, contrasting favorably with the convolutional methods (c.f. Figure 9), which often yield partial or malformed trees when interpolations are attempted. These results underscore the benefits of a continuous geometric representation combined with a recursive, topology-aware decoder.

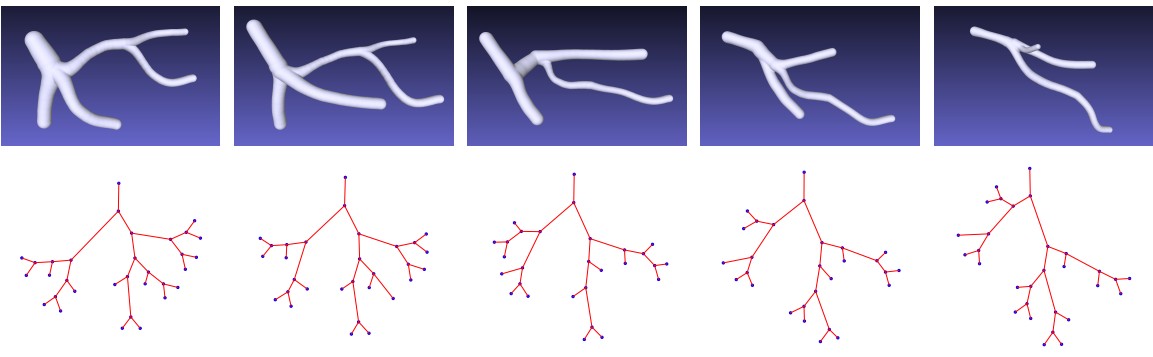

Figure 3: (First row) Interpolations produced by the proposed 3D vessel tree autoencoder between two VesselTrees validation set examples. (Second row) Interpolations produced by the proposed 2D tree autoencoder.

## 6. Conclusion

In this paper, we introduced a two-stage Transformer-based framework that learns vector representations of tree-structured anatomical data by separately encoding continuous vessel geometry and overall vascular topology. Through sequential training of the Vessel Autoencoder and Vessel Tree Autoencoder, our method achieves high reconstruction fidelity while ensuring topological consistency via a recursive decoding scheme. Notably, the approach requires significantly less GPU memory than 3D convolutional baselines, making large-batch training feasible. Experiments on a synthetic 2D tree dataset and a 3D coronary artery dataset demonstrate superior reconstruction accuracy and topology preservation. Furthermore, we demonstrate that our framework can produce smooth and plausible latent-space interpolations. Overall, these findings underscore the promise of vector encodings for scalable, precise, and topologically reliable modeling of complex anatomical trees. Future work will include ablation studies to rigorously assess individual model components and evaluations of anatomical fidelity, further enhancing clinical applicability.

## Acknowledgments

This research was funded by HeartFlow, Inc.; James Batten was supported by the UKRI CDT in AI for Healthcare (Grant No. EP/S023283/1).

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

## Appendix A. Vessel Autoencoder

### A.1. Gaussian-Smoothed Curvature

The gaussian-smoothed curvature is computed in the data pre-processing stage. For each vessel of shape $(n, 4)$, we compute a curvature term using:

Given a sequence $\mathbf{P}(i)$, $\quad i = 1, \ldots, n$,

(1) Compute the first derivative: $\mathbf{D}(i) = \mathbf{P}(i+1) - \mathbf{P}(i)$, $\hfill (5)$

(2) Smooth $\mathbf{D}$ with a Gaussian filter: $\widetilde{\mathbf{D}}(i) = G_\sigma * \mathbf{D}(i)$, $\hfill (6)$

(3) Compute the second derivative of the smoothed data: $\mathbf{D}^{(2)}(i) = \widetilde{\mathbf{D}}(i+1) - \widetilde{\mathbf{D}}(i)$, $\hfill (7)$

(4) Define the smoothed curvature: $c_\sigma(i) = \left\| \mathbf{D}^{(2)}(i) \right\|$. $\hfill (8)$

### A.2. Vessel Segments

We segment the vessel into 64 partitions according to the compute_segments algorithm (c.f. Algorithm 4) using a sensitivity of 0.75. During training, within each segment, a 5-dim (x, y, z, r, t) value is sampled. The resulting set of samples of shape (64, 5) is then passed to the Vessel Encoder.

### A.3. Sinusoidal features

For the x, y, z terms, we lift these to sinusoidal "fourier features", using octaves [1, 2, 4, 8, 16, 32] according to Eq. 18. This lifting is performed to the values which are passed both to the encoder and the decoder (x, y, z, t). Note that on the output of the decoder, the values are predicted in the original Euclidean domain. We find that this lifting technique significantly improves the reconstruction accuracy of the Vessel Autoencoder.

### A.4. Loss function reweighting

In our initial experiments training the Vessel Autoencoder, we observed that the reconstruction accuracy was lower for longer and more tortuous vessels, despite the fact that all vessels are normalized to the same domain during training. In order to remedy this, we re-weight the loss function to increase the importance of these cases. For each vessel, we compute the euclidean distance $d_e$ between the endpoints in the original domain, in addition to the arc length of the vessel $d_a$. The MSE loss for each vessel in the training batch is then multiplied by:

$$\alpha \cdot \frac{d_a}{\sqrt{d_e}} \qquad (9)$$

In our experiments we set $\alpha$ to 0.3. Furthermore, we re-weight the MSE loss to encourage the model to focus more on the position terms $x, y, z$ and less on the radius terms $r$. We compute these separately and multiply the position and radius loss terms by 1.0 and 0.01 respectively.

### A.5. Vessel Autoencoder Architecture

**Vessel Encoder.** The Vessel Encoder is composed of a Transformer Encoder and two 2-layer MLPs. Both MLPs have a hidden dimension of size 2048 and use a GELU activation. The first MLP takes as input the set of (x, y, z, r, t) values (lifted to sinusoidsal features) and outputs a set of vectors of dimension 512. These are then passed through a Transformer Encoder with 6 layers and 12 heads. The resulting features are pooled over the set dimension into a single vector of size 512, which is passed through the second MLP to produce the vector representation of the vessel of size 64.

    **Vessel Decoder.** The Vessel Decoder is composed of a single 2-layer MLP using a GELU activation. It takes as input the vector representation of the vessel $z_v$ and a scalar value $t$ (lifted to sinusoidal features) concatenated over the feature dimension and outputs the 4-dim vector $f_d(z_v, t)$. We then interpolate between the vessel endpoints a and b using Formula 1 to calculate $g_d(z_v, t)$.

### A.6. Matching endpoints

During training we set $m(t) = 1$ as shown in Formula 1. During evaluation, we use $m(t) = 0.5^{-0.2} \cdot t^{0.1}(1-t)^{0.1}$. Using this function has the benefit of guaranteeing that the endpoints of the reconstructed vessel match $p_a$ and $p_b$ since $m(0) = 0$ and $m(1) = 0$ (and that the endpoint radii match $r_a$ and $r_b$). Using $m(t) = 1$ during training results in better stability.

## Appendix B. Vessel Tree Autoencoder

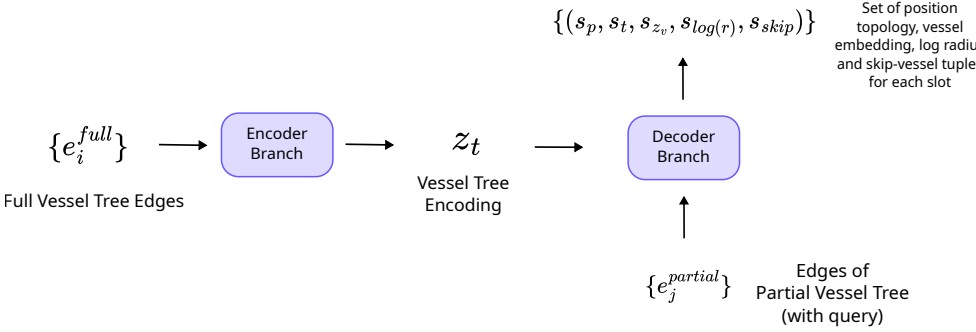

Figure 4: Diagram of the Vessel Tree autoencoder model. The encoder branch outputs a global vector $z_t$, and the decoder branch uses fixed learnable slots (via a Transformer Decoder) to predict child nodes from the partial tree, conditioned on $z_t$.

### B.1. Vessel Tree Autoencoder: Edge features and Encoder Branch

**Input edge features**. We describe the full input tree connectivity structure as a set of directed edges $\{e_i^{\text{full}}\}$. Concatenating the (positional and topological) features of the outgoing node $n_a$ and the incoming node $n_b$ creates each directed edge (c.f. Figure 5).

These features include the topological vectors $t_a$ and $t_b$, as well as the positional information $p_a$ and $p_b$ (in practice, we raise the 2D or 3D scalar coordinates to "fourier feature" sinusoidal embeddings) (Tancik et al., 2020). These topological terms $t_a$ and $t_b$ are one-hot vectors that represent the number of children of the corresponding nodes. For instance, the one-hot representation of a leaf node node $n_{\text{leaf}}$ would be $(0, 0, 1)$, whereas the one-hot representation of a bifurcation node $n_{\text{bifurcation}}$ would be $(1, 0, 0)$. Note that, in principle, these topological features can be removed from the edge representations, since the connectivity structure can be inferred from the edge positional information alone. We found, however, that including these features leads to improved performance with the proposed model.

Additionally, as mentioned above, these input edge features also contain an optional query flag (which is a binary scalar value equal either to 0.0 or 1.0). This query value is absent from the full tree edge representations (input to the encoder branch) $\{e_i^{\text{full}}\}$ and is only used in the partial tree edges $\{e_j^{\text{partial}}\}$.

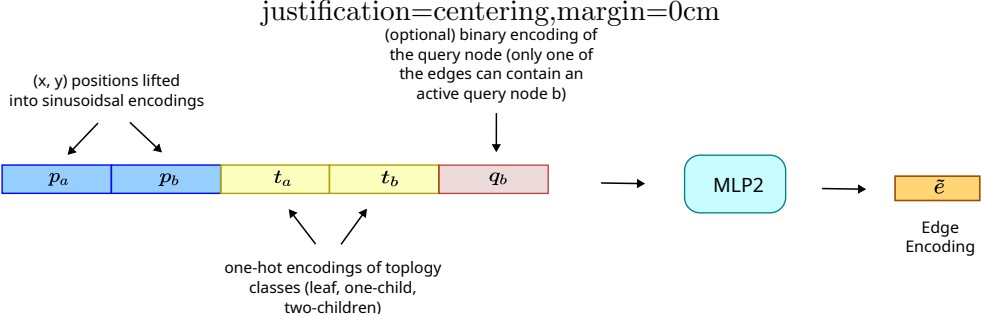

Figure 5: Diagram of the Single Edge Encoder in the 2D case. On the 3D VesselTrees dataset, we concatenate the radius terms of the two endpoints $r_a$ and $r_b$, in addition to the vessel embedding $z_v$ to this vector.

This set of edge vector encodings $\{\tilde{e}_i\}$ is a sufficiently complete representation of the input tree geometry to enable the encoder branch to represent both the discrete connectivity graph and (in the 3D case) the continuous structure as a single vector. We demonstrate this empirically by adding a decoder to this model and training the resulting autoencoder to reconstruct the original tree from the vector representation of the tree $z_t$.

**Single Edge Encoder**. A two-layer MLP with a GELU activation makes up the initial part of the encoder branch, which we refer to as the Single Edge Encoder (c.f. Eq 10). The edge features from the input set ($\{e_i^{\text{full}}\}$ or $\{e_j^{\text{partial}}\}$) are consumed by this MLP, which lifts them to an edge embedding $\tilde{e}$ with a feature dimension of size $64 \times n_{heads}$. In this and the following sections we write $n_{heads}$ as the number of heads in the transformer encoder and decoders).

$$\tilde{e}_i = \text{SingleEdgeEncoder}(e_i) \tag{10}$$

**Edges Encoder**. We pass the resulting set $\{\tilde{e}_i\}$ through a Transformer Encoder and to produce a set of vectors $\{\hat{e}_i\}$. Note that the vector in the sets $\{\tilde{e}_i\}$ and $\{\hat{e}_i\}$ are of the same

size $64 \times n_{heads}$. The Edges Encoder is the name of the component which includes both the Single Edge Encoder and the Transformer Encoder (c.f. Eq. 11 and Fig. 6). Since the same edge encoding scheme is used in the encoder and decoder branches, we use the same notation here for these edges. On the SSA dataset we use a Transformer Encoder with 6 layers and 12 heads. On the VesselTrees dataset we use a Transformer Encoder with 12 layers and 16 heads.

$$\{\hat{e}_i\} = \text{EdgesEncoder}(\{\tilde{e}_i\}) \tag{11}$$

A key benefit of processing lifted edge encodings using the transformer encoder architecture is that it preserves permutation equivariance with respect to the input set. Since the directed edges do not possess a canonical ordering, this property enables the transformer to generalise its feature representations across arbitrary permutations of the input set, instead of considering different permutations of these edges as independent samples in the data distribution.

justification=centering,margin=0cm

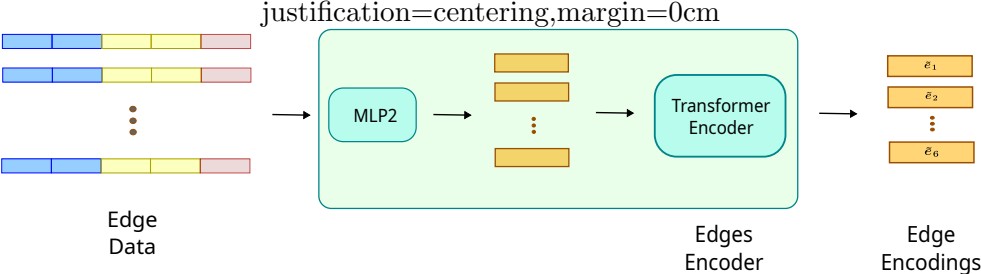

Figure 6: Diagram of the Edges Encoder

**Pooling Head**. We include a pooling head at the end of the encoder pathway that first applies a two-layer MLP to map the edge encodings $\{\hat{e}_i\}$ to the target embedding size $z_{dim}$ before taking the mean of the resulting representations over the edge dimension (c.f. Figure 7).

$$z_t = \text{mean}(\text{MLP2}(\{\hat{e}_i^{\text{full}}\})) \tag{12}$$

As an additional implementation detail, we include an "edge mask" for both the full and partial trees. This edge mask indicates which edges are "active" in the input set. The pooling described in this section is performed with respect to this mask (only the active edge features are included in the final vector representation).

## B.2. Partial Tree Encoder

**Start token**. The decoder branch must be capable of accepting empty partial tree representations as input in order to correctly perform the first step in the tree reconstruction process. We add a "start token" to the set of partial edge encodings $\{e_j^{partial}\}$. This start

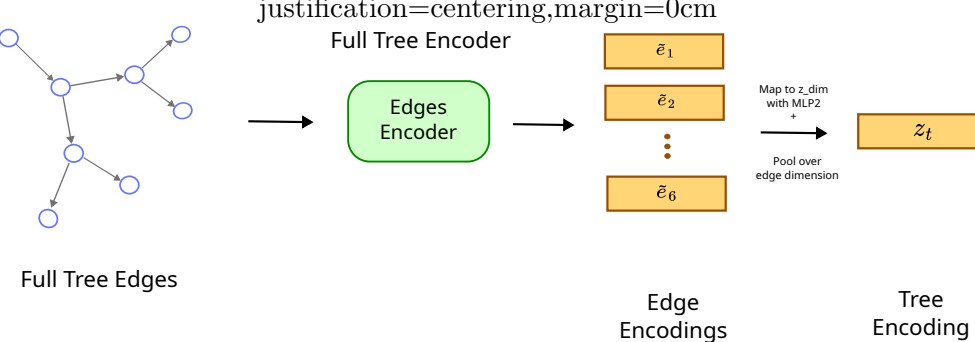

Figure 7: Diagram of the Encoder Branch. The Edges Encoder in this branch is the Full Tree Encoder, which is followed by a pooling operation to obtain the vector representation of the full tree $z_t$.

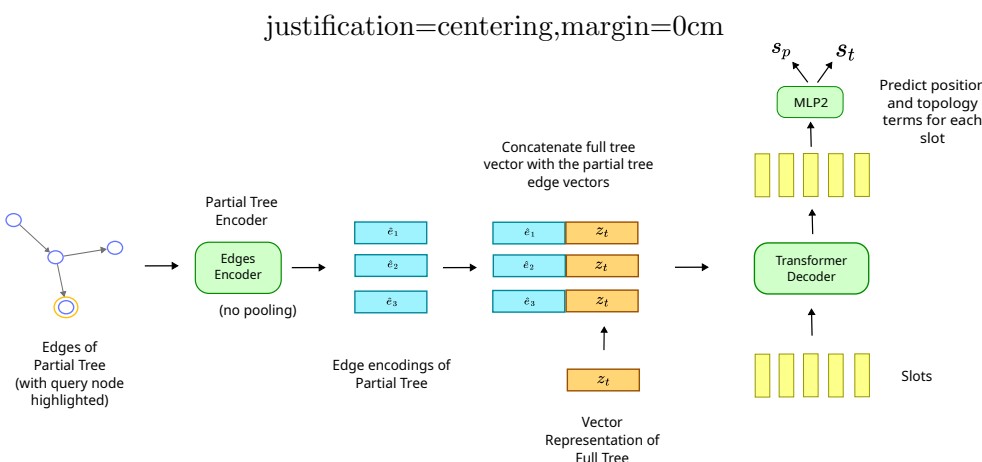

Figure 8: Diagram of the Decoder Branch. The Edges Encoder in this branch is the Partial Tree Encoder, which is not followed by a pooling operation. The vector representation of the full tree $z_t$ is concatenated along the feature dimension of each of the partial tree edge encoding. This set of concatenated vectors is then passed as context to the Transformer Decoder which predicts properties for each slot vector. Note that this example illustrates the slot predictions in the 2D case. In the 3D case, there are additional properties that are predicted for each slot: the log radius, the vessel embedding and the skip-vessel flag

token is implemented as a learnable vector of size $(64 \times n_{heads})$ initialized from a uniform random distribution in the interval $[-1.0, 1.0]$. When the partial tree is empty at the start of the decoding process, we have: $\{e_j^{partial}\} = \{\text{token}_{start}\}$.

Many sequence-to-sequence learning models, including BERT (Devlin, 2018), GPT (Radford, 2018), and the original Transformer (Vaswani, 2017), use unique tokens like CLS to indicate the start of a sequence. These tokens aid the models in distinguishing the initial state from the other states. The presence of the "start token" enables the model to be prompted with an empty set. As such, in the sitution where $\{e_j^{partial}\} = \{\text{token}_{start}\}$, the decoder model is in the first state of the recursive decoding process and is tasked with decoding the root node properties from the full tree vector representation $z_t$.

**Semi Edge**. After the initial decoding step the partial tree is exclusively composed of the root node. However, since the decoder expects a set of input *edges*, it's necessary to introduce another technique enabling the model to correctly handle this situation. In the VeTTA model, we propose a straightforward addition to the partial tree representation that we refer to as the "Semi Edge" to get around this problem. This semi-edge connects the root node to itself, and is included in both the full and partial tree edges. After the first step in the decoder process, we simply add the semi edge $e_{semi}$ to this set:

$$\{e_j^{partial}\} = \{\text{token}_{start}, e_{semi}\}, \text{ where } e_{semi} = (n_{root}, n_{root}) \tag{13}$$

Although the concept of incorportating "self-loops" has been studied in the field of deep geometric learning (Veličković et al., 2017) the "Semi Edge" presents a fresh application of this idea to the tree decoding problem. Thus, with this new edge, the model can distinguish between the empty partial tree (which only has the start token), and the partial tree containing only the root node (corresponding to the start token and the semi edge).

**Non-proximal edge filtering**. The partial tree decoder method leverages another technique that involves filtering out edges that do not belong to the sequence of nodes connecting the root node to the query node. We call these the "non-proximal" edges in the partial tree representation (since they are not proximal to the query node). This filtering procedure involves setting the corresponding edge masks to zero. In our experiments, we found that this filtering improves the stability of the optimization during training.

**Filtering additional information in the partial tree**. Additionally, we found in our experiments that removing the vessel embeddings and the radius terms before passing them to the partial tree encoder improves reconstruction accuracy on the 3D experiments. We hypothesize that this may be due to errors which can compound in the recursive decoding. By removing this information from the edges before they are fed into the partial tree encoder, we mitigate this behavior. In terms of implementation this simply involves not concatenating these properties along the feature dimension of the edges $\{e_j^{partial}\}$ (c.f. Fig. 5).

**Skip-vessel flag**. In our experiments we found that trying to include vessels smaller than 2.5mm into the training set of the Vessel Autoencoder caused a number of issues such as making training more unstable. In the VesselTrees dataset, trifurcations are typically represented by two bifurcations close to one another. In order to solve this problem we introduce an explicit binary "skip-vessel" flag into the edge-vector representations on the input to the recursive model, and include this as a feature predicted by the slot-vectors on

the output of the model. For the steps in the recursive decoding where the ground-truth child node skip-vessel flag is set to 1, we exclude the vessel embedding terms from the cost-matrix. This has the effect of ignoring the embeddings predicted by the recursive decoder for these targets. During evaluation, if a skip-vessel flag of value 1 is predicted by the decoder, then for the corresponding vessel we decode a linearly-interpolated vessel between the endpoint properties $(p_a, r_a)$ and $(p_b, r_b)$.

**Injecting the Full Tree Vector**. In our approach, we concatenate the full tree embedding $z_t$ to each of the partial tree edge encodings $\{\hat{e}_i^{\text{partial}}\}$ (on the final layer of the edges encoder in the decoder branch) in order to condition the decoder branch on the vector representation of the tree produced by the encoder pathway (c.f. Figure 8).

**Slots-based Decoder**. The last component of the decoder branch binds a set of slots to the query's child nodes. To do this, we use a Transformer Decoder, performs a layer-by-layer updating of a set of learnt high-dimensional slot vectors, conditioned on the encoded partial tree edges $\{\tilde{e}_j^{partial}\}$. We train an unordered set of vector "slots", which are then processed by the transformer decoder. We define this set of slots as:

$$S = \{s_i \in \mathbb{R}^{D_{\text{slots}}} \mid 1 \le i \le M_{\text{slots}}\} \tag{14}$$

Where $M_{\text{slots}} = 32$ represents the number of slots, and $D_{\text{slots}} = 64 \times n_{\text{heads}}$ is the feature dimension of these vectors. This approach is strongly related to Slot Attention (Locatello et al., 2020).

**MLP Predictor Head**. We add a two-layer MLP with a GELU activation at the end of the decoder branch (c.f. Figure 8). This MLP serves to transform the slot embeddings on the transformer decoder's output $\{s_i^{l_{\max}}\}$ into the properties of the target objects. Applying the MLP to each slot in $\{s_i^{l_{\max}}\}$ results in the transformed slots $S_{\text{prop}}$:

$$S_{\text{prop}} = \{\text{MLP}(s_i^{l_{\max}}) \mid 1 \le i \le M_{\text{slots}}\} \tag{15}$$

**Set Prediction**. Each recursive step in the tree decoding process can be described as a *set prediction* problem. To decode the properties of the child nodes, we first partition the slots $S_{\text{transformed}}$ into groups using a clustering algorithm.

$$\text{Cluster}(S_{\text{prop}}) \rightarrow \{C_1, C_2, \dots, C_K\} \tag{16}$$

Where $C_k$ represents the $k$-th cluster of slots, and $K$ is the number of clusters. This number, which should equal the number of child nodes, is always predicted by the topology term in the previous recursive step. In this paper, the maximum number of clusters is 2, however this method also works with target sets of cardinality greater than 2.

Then, by aggregating the slots in each cluster, we predict each child node's attributes:

$$p_k = \text{Aggregate}(C_k), \quad 1 \le k \le K \tag{17}$$

The function Aggregate in our implementation simply averages the slot characteristics inside each cluster to obtain a single vector per cluster. In practice, we find that the "average" connecting agglomerative clustering algorithm performs well on this task.

**Clustering in the lifted domain**. As mentioned previously, we lift the positional coordinates $p_a$ and $p_b$ in the edge features to the fourier/sinusoidal domain. To do this, we

use the sine and cosine functions specified in Eq. 18 with various octaves (alpha) for each coordinate.

$$F(\alpha, x, y) = (\cos(2\pi\alpha x), \sin(2\pi\alpha x), \cos(2\pi\alpha y), \sin(2\pi\alpha y)). \tag{18}$$

In addition, the model regresses the slot properties on the VeTTA decoder's output in the same lifted domain as the inputs. We concatenate these positional Fourier features with the one-hot topology vectors to produce the final co-domain. In 3D we also concatenate the log radius, the vessel embedding $z_v$ and the skip-vessel flag to this representation. We write the lifted targets at each recursive step as $\Phi(g_j)$, where $\Phi$ is the lifting function described above.

In order to perform the clustering, we compute the pairwise distances between predicted slots and target nodes in the lifted domain, as illustrated in Equation 19:

$$C_{ij} = \|s_i - \Phi(g_j)\|_2, \tag{19}$$

Where $i$ and $j$ are indices of the predicted slots and target nodes, respectively, and $C_{ij}$ denotes an entry in the cost/distance matrix. We then use this distance matrix $C$ to compute the agglomerative clustering. The key idea here is that performing this clustering operation in the lifted co-domain simultaneously resolves multiple properties of the slots into a small number of set-structured predictions (where each prediction is a vector describing the properties of one element of the output set). In particular, we find that performing this type of clustering of positional information in the sinusoidal domain significantly outperforms the same approach applied in the original spatial domain. In, since 3D these properties include the vector embedding of the vessel $z_v$, we can describe one aspect of this operation as a clustering over continuous trajectories.

**Inverting the lifting function**. While the model takes as input and outputs features in the lifted domain during training, a location in the original Euclidean domain corresponding to this high-dimensional feature must be computed in order to interpret these outputs during evaluation. We simply generate a dense sequence of 1000 linearly-spaced values in the predefined range (in our case, we choose $[-3.0, 3.0]$) and select the coordinate that minimises the distance in the sinusoidal domain in our implementation rather than offering a closed form solution to the minimization problem (c.f. Eq. 20). The closest matching coordinate along each axis is found separately rather than creating a dense 2D (or 3D) grid because the lifting formula 18 remains identical along each cartesian axis.

$$(x, y) = \underset{x, y \in \mathbb{R}^2}{\arg\min} \|F(\alpha, x, y) - F(\alpha, x_p, y_p)\|_2, \tag{20}$$

### B.3. Loss Function

**Right hand matching**. Given a cost matrix and a target mask, the right-hand matching algorithm (c.f. Algorithm 3) seeks to identify two minimum-cost directed matchings between a set of predictions and targets. The active members in the target set are indicated by the target mask, a binary tensor. The algorithm initially determines active targets and generates an active cost matrix. The Hungarian matching technique is then used to calculate the best assignment between predictions and active targets. The

matching matrix R is then updated by setting the slot matched to each of the active targets and returned.

**Top-k Matching**. Given a cost matrix, target mask, and integer k, the Top-K Matching Algorithm (c.f. Algorithm 2) is devised to identify the top-k minimum-cost directed matches between a collection of predictions and targets. The procedure begins by figuring out how many targets there are and initializing the L and R matching matrices. The right_hand_matching algorithm is then executed k times, updating the matching matrices L and R as well as the cost matrix copy C' every cycle. The algorithm determines the remaining unmatched predictions after k iterations and modifies the cost matrix copy C' to take into account only active targets. The remaining predictions are then given the nearest target indices, and the matching matrix L is subsequently updated. Two matching matrices, L and R are the algorithm's final output.

### B.4. Augmentation

We employ two types of augmentation in the proposed method to prevent overfitting: global transformations of the tree and local jitter.

**Global Augmentations**. Three categories of global augmentation – translation, rotation, and zooming – are used for the 2D and 3D data. In 2D we sample rotations from a uniform distribution in the range $[-45.0, 45.0]$ degrees, and in 3D we use fully random 3D rotations. Zoom factors are sampled in the range $[0.75, 1.5]$.

**Local Jitter**. Due to the recursive nature of the decoding, inconsistencies can quickly compound and lead the prediction to deviate significantly from the original structure. We add a slight gaussian "jitter" augmentation (with standard deviation $\sigma = 0.005$) to the nodes in the partial tree in order to reduce this behavior.

## Appendix C. Interpolation Experiments on the SSA dataset

In Figure 9, we show interpolations between the two test set examples we use to produce the 2D interpolations in Figure 3 with the proposed model. Here we show the reconstructions for the latent codes interpolated at corresponding steps for the two convolutional baseline models, using group and batch normalization (Conv-2D-VAE (GN) and Conv-2D-VAE (BN) respectively). Note that this figure also illustrates the poor reconstruction behavior of the batch norm + ReLU model in the variational setting compared to the model trained using group norm + GELU.

We emphasize here that the reporting the reconstruction accuracy on the SSA dataset is not the main focus of the paper, since this is a toy dataset and is not meant to be representative of real-world vasculature. The convolutional baseline Conv-2D-VAE (GN) achieves almost perfect reconstructions of the input segmentations in a variational setting, and the numbers we report in the results table primarily serve to show that the reconstruction accuracy of our method is on-par with this baseline. The interpolation experiments shown in the above figure better illustrate what we seek to demonstrate in this paper using this dataset: that convolutional methods fail to learn topologically-sound latent spaces, and interpolations between validation samples consistently fail to reconstruct plausible trees. Our method, however, can be empirically shown to produce highly plausible trees with correct topology throughout the latent domain (c.f. Fig. 3).

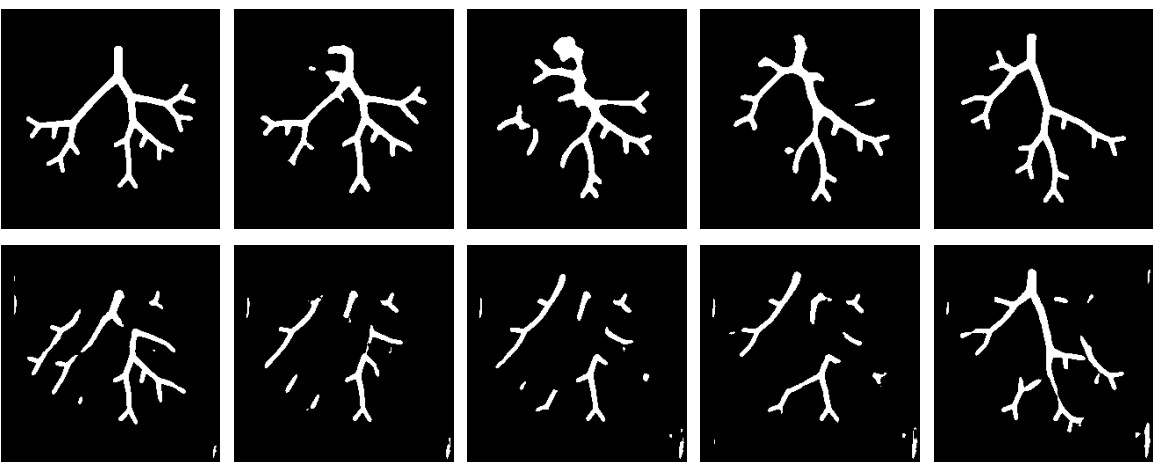

Figure 9: (First row) Interpolations produced using the baseline model Conv-2D-VAE (GN). (Second row) Interpolations produced using the baseline model Conv-2D-VAE (BN).

## Appendix D. Baseline Model Architectures and Training

### D.1. Conv-2D (GN) and Conv-2D (BN)

The Conv-2D models have three different levels in the encoder and the decoder, with channels $[1, 32, 64, 128]$. At each level we use 4 residual layers. The models marked with (GN) use group normalization and GELU activations, the models marked with (BN) use batch normalization and ReLU activations. We train these models with a batch size of 128, linearly ramping up the learning rate to $3e-4$ over 1000 steps, and then decaying by a factor 10 every 20k steps. All the models are trained for 34k steps. These convolutional models are trained using translation, zooming and rotation augmentations.

### D.2. GDVM

We adapt the architecture proposed in (Brock et al., 2016) to volumes of size $128^3$. In our implementation, which we refer to as GDVM in this paper, to make it compatible with volumes of this size with minimal changes, we take the architecture proposed by the authors and increase the number of units in the dense layer enc_fc1 from 343 to 32768, and increase the bottleneck dimension from 512 to 8192.

We train the resulting autoencoder in both a standard and variational configuration with a batch size of 16 for 80k steps. We use a linear ramp up of the learning rate over the first 1000 steps to reach a peak learning rate of $3e-4$, then we decay the learning rate by a factor 10 every 40k steps. The models are trained using full 3D rotation, translation and zooming augmentations.

The original code for the GDVM model can be found at https://github.com/ajbrock/Generative-and-Discriminative-Voxel-Modeling/blob/master/Generative/VAE.py.

### D.3. Conv-3D

The Conv-3D model are adapted from the GDVM architecture with some more modern techniques. Since we observe in the 2D experiments that using group normalisation and GELU leads to more stable performance in both configurations, we use that in this model (replacing batch norm and ELU activations in the GDVM model). We increase the channels in the encoder from $[1, 8, 16, 32, 64]$ to $[1, 16, 64, 256, 1024]$ and add a residual layer at each level. We apply the same changes to the decoder. We find that these changes substantially improve the reconstruction accuracy of the resulting model in both the standard and variational configurations.

We train these autoencoders with a batch size of 16 for 40k steps. We use a linear ramp up of the learning rate over the first 1000 steps to reach a peak learning rate of $3e-4$, then we decay the learning rate by a factor 10 every 20k steps. The models are trained using full 3D rotation, translation and zooming augmentations.

### D.4. VesselVAE

We take the architecture proposed in (Feldman et al., 2023), and adapt our data to work with their code. We linearly transform the VesselTrees dataset to match the statistics of their preprocessed data and set the root node to be always located at the origin. In addition, we resample the number of nodes down each vessel such that these statistics also match. We train with a batch size of 10 for 23k steps with a constant learning rate of $1e-4$, with the goal of replicating their methodology to the best extent. One of the changes we make is to increase the bottleneck dimension to 8192 in order to maintain an apples-to-apples comparison with respect to the other models.

It should be well noted that the VesselVAE paper is positioned as a *generative* method, where the authors clearly state that their objective is to use the resulting model to sample novel vessel geometries, and the evaluations they perform are based on sampled tree statistics. As such, comparing against this method on reconstruction metrics is somewhat unfair since it is not the intended goal of their approach. We include it in the results table since it is the model in the literature which bears the greatest similarity to our method. Another point of note for the VesselVAE method is that it is trained on IntrA (Yang et al., 2020) dataset, which is generated from 103 models of brain vessels. This dataset is on a much smaller scale than the VesselTrees dataset, and is likely less suitable for the training of larger models.

We hypothesize that the encoder in their model does not retain significant aspects of the input structure in the latent embedding, which explains the degraded performance of this method compared to the other models in the results table. In order to test this hypothesis, we downloaded the checkpoint they provided on their HuggingFace demo and provided it with one of the preprocessed trees from their dataset. We observe that the mesh produced on the output of their model has significant differences compared to the original mesh (c.f. Fig. 10), which supports this hypothesis.

For more details, we refer the reader to the VesselVAE GitHub repository at https://github.com/LIA-DiTella/VesselVAE and the HuggingFace demo at https://huggingface.co/spaces/paufeldman/vv.

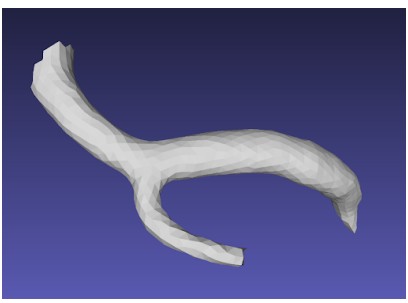 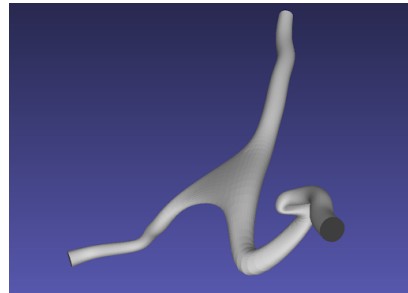

Figure 10: (Left) Original mesh of the preprocessed sample passed as input to the VesselVAE encoder. (Right) Reconstructed mesh taken from the output of the VesselVAE decoder.

## Appendix E. Algorithms

---
**Algorithm 1:** Loss Calculation Algorithm

---
**Input:** Cost matrix $C$ of shape $(s, t)$, target mask $M$ of shape $(t)$, integer $k$

**Output:** Scalar loss value

L, R $\leftarrow$ **top_k_matching**$(C, M, k)$;

loss_lhs $\leftarrow$ **sum**$(C \cdot L)/$**sum**$(L)$ ;                              // Compute loss lhs

loss_rhs $\leftarrow$ **sum**$(C \cdot R)/$**sum**$(R)$ ;                              // Compute loss rhs

loss $\leftarrow$ loss_lhs + loss_rhs ;                              // Compute final loss

**return** *loss*;

---

---
**Algorithm 2:** Top-K Matching Algorithm

---
**Input:** Cost matrix $C$ of shape $(s, t)$, target mask $M$ of shape $(t)$, integer $k$
**Output:** Matching matrices $L$ and $R$ of shape $(s, t)$
$N \leftarrow \sum M$;                                                         // Number of targets
$C' \leftarrow \textbf{copy}(C)$;                                             // Copy of cost matrix
$\text{max\_value} \leftarrow \max(C) + 1.0$;                                 // Maximum value for update
$L \leftarrow \textbf{zeros\_like}(C)$;                                       // Initialize matching matrices
$R \leftarrow \textbf{zeros\_like}(C)$
**for** $i \leftarrow 0$ **to** $k - 1$ **do**                                // Loop from 0 to $k - 1$
$\quad\big|\quad$ _, $R_i \leftarrow \textbf{right\_hand\_matching}(C', M)$;  // Matching RHS
$\quad\big|\quad L \leftarrow L + R_i$;                                       // Update matching accumulators
$\quad\big|\quad R \leftarrow R + R_i$
$\quad\big|\quad P \leftarrow \textbf{argwhere}(R_i > 0.5)[:, 0]$;            // Matched prediction indices
$\quad\big|\quad C'[P, :] \leftarrow \text{max\_value}$;                      // Update cost matrix copy
**end**
$P \leftarrow \textbf{argwhere}\big(\textbf{sum}(L, \text{axis} = 1) < 0.5\big)[:, 0]$;   // Remaining prediction indices
$C' \leftarrow C' + \text{max\_value} \times \textbf{reshape}(1.0 - M, (1, -1))$;   // Update cost matrix copy
$T \leftarrow \textbf{argmin}(C'[P, :], \text{axis} = 1)$;                    // Closest target indices
$L[P, T] \leftarrow 1.0$;                                                     // Update matching matrix $L$
**return** $L$, $R$;

---

---
**Algorithm 3:** Right Hand Matching Algorithm

---
**Input:** Cost matrix $C$ of shape $(s, t)$, target mask $M$ of shape $(t)$
**Output:** Matching matrix $R$ of shape $(s, t)$
$s \leftarrow C.\text{shape}[0]$;                                             // Number of predictions
$t \leftarrow C.\text{shape}[1]$;                                             // Number of targets
$\text{active\_tgt\_idxs} \leftarrow \textbf{argwhere}(M > 0.5)$;             // Active target indices
$C_{\text{active}} \leftarrow C[:, \text{active\_tgt\_idxs}[:, 0]]$;          // Active cost matrix
$\text{row\_ind}, \text{col\_ind} \leftarrow \textbf{linear\_sum\_assignment}(C_{\text{active}})$;   // Bipartite matching
$R \leftarrow \textbf{zeros}(s, t)$
$R[\text{row\_ind}, \text{active\_tgt\_idxs}[\text{col\_ind}, 0]] \leftarrow 1$;   // Update matching $R$
**return** $R$;

---

---

**Algorithm 4:** Compute Segments Algorithm

---

**Input:** curvature: 1D array of length $n$, integer n_segments, float sensitivity

**Output:** segments: List of (start_index, end_index) pairs

minval $\leftarrow$ min(curvature);

maxval $\leftarrow$ max(curvature);

tcurvature $\leftarrow \frac{\text{curvature}-\text{minval}}{\text{maxval}-\text{minval}}$;

tcurvature $\leftarrow$ (tcurvature)$^{\text{sensitivity}}$;

ccurvature $\leftarrow$ cumsum(tcurvature);

segment_boundaries $\leftarrow$ linspace$\big(0,\ \max(\text{ccurvature}),\ \text{n\_segments}+1\big)$;

segments $\leftarrow$ [];

last_index $\leftarrow 0$;

**for** $i \leftarrow 0$ **to** $n\_segments - 1$ **do**

  indices $\leftarrow$ argwhere$\big($ccurvature $\geq$ segment_boundaries$[i]\ \wedge$ ccurvature $<$

   segment_boundaries$[i+1]\big)$;

  **if** $|indices| > 0$ **then**

    index_0 $\leftarrow$ max(indices[0], last_index);

    **if** $i < n\_segments - 1$ **then**

      index_1 $\leftarrow$ max(indices[−1] + 1, index_0 + 1);

      append (index_0, index_1) to segments;

    **end**

    **else**

      append (index_0, $n$) to segments;

    **end**

    last_index $\leftarrow$ segments[−1][1];

  **end**

  **else**

    append (last_index, last_index + 1) to segments;

    last_index $\leftarrow$ last_index + 1;

  **end**

**end**

**return** *segments*;

---

