# OpenReview forum: "Vector Representations of Vessel Trees"
_MIDL.io/2025/Conference — MIDL 2025 Oral_

### Official Review · Reviewer_ctVV · 2025-02-19

**Confidence:** 3
**Preliminary Rating:** 5
**Recommendation:** Oral
**Final Rating:** 5

**Summary:**

This work introduce a framework that learn a vector representation of graph datas with an application on vascular networks. This framework consists of a first encoder dedicated to vessels and a second dedicated to the entire tree. Finally, the decoder reconstructs the tree by iteratively predicting the children of each node starting from the root node.

The authors evaluated their framework on a public 2D synthetic tree dataset and in house 3D coronary artery dataset. They compare the proposed method to autoencoder baselines and other representation learning methods for vessel like structures. The conducted experiments demonstrate the good performance of the proposed framework.

Currently, vessels are mostly model through their binary segmentation masks, this representation is not ideal to capture topological properties. Thus, the analysis of vessels trees through their graph representation is an exciting line of research to better take into account geometric and topological properties. This work is a very interesting contribution to this field of research.

**Strengths:**

This paper proposes a novel representation method for vessel trees. The novelty of the method lies in the use of the graph representation instead of the binary segmentation map representation. The authors leverage transformers cleverly to encode vessel trees. Also, the iterative design of the decoder allows to constraint the reconstruction to be a valid tree structure.

The goal of the paper seems relevant with respect to the state-of-the-art.

The experiments are motivated and clearly demonstrate the interest of the method.

**Weaknesses:**

During my reading of the paper, I had some difficulties understanding some technical details of the presented method. For example, the notation used are not systematically introduced. I think the presented method is dense, and It was not easy for the authors to detail the whole method (in addition, the related works help a lot) . However, I have the feeling that some adjustment can be done in the main paper.

**Detailed Comments:**

Overall, I really found this paper innovative and relevant regarding the state-of-the-art. As the presented method is quite dense, It can be difficult for the reader to understand clearly with only the main paper, but the supplementary materials add a lot of information.

To me, this paper would be a very strong contribution to MIDL and for the community. However, I think the clarity of the paper can be improved in the rebuttal.

I listed in the questions sections some of my interrogations which could be shared by other readers. I also made some suggestions for improved readability.

I don't expect the authors to answer all my comments, but I hope these comments can be useful feedback for them.

**Justification Of The Final Rating:**

This is very nice work with clear novelty and a well written paper.  The authors did their best to improve the paper readability and I have the feeling that the reviewers comments helped to did so. Moreover, the supplementary materials are also well written and exhaustive. I agree with the authors that the supplementary materials provide necessary explanation about the method and therefore must be keep.

**Justification Of The Preliminary Rating:**

Overall, the presented method is clearly innovative and demonstrate good performance. It fills a gap in an area of research with high potential. Moreover, the evaluation part is solid and the scientific approach rigorous. The only small limitation is the paper readability.

For these reasons, I choose to rate this paper as a strong accept.

**Questions To Address In The Rebuttal:**

### Major remarks


1 - If I understood correctly the authors used Fourier features to encodes the input features $(x, y, z, r, t)$. This technical choice is not motivated, so I would like some insights about it.

2 - The notation at the beginning of the Method section ($n$ and $(x, y, z, r, t)$) are not clearly introduced. As it describes the input data of the model I think the clarity of the paper would benefit of such explanation.

3 - For the Vessel Tree Autoencoder the input data is the set of edges representing the vessel tree. The authors say edges features are the nodes coordinates and topological attributes. However, these topological attributes are described only in the supplementary materials. I think this is misleading because the term "topology" can refer in the literature to various subjects (actual topological descriptors such as betti numbers or geometric descriptors such as curvature or vessel lenght for example). In this case, it refers to the number of children. I think It needs to be specified in the main paper.

4 - Also the edges features comprise the $z_v$ from the vessel encoder but only in 3D setting. If I'm correct It can also be done in the 2D setting, so I would like to know if the authors had a motivation behind this design choice.

5 - If I'm not mistaken, the evaluation currently only involve baselines using binary segmentation representation. I was wondering if others methods using the graph representation exist in the literature. If so, It must be included in the current evaluation.

6 - Finally, as the method is complex and implies numerous design choice, I think an ablation study would be very beneficial. Mainly, it seems to me that the *Vessel autoencoder* is maybe unnecessary, as It is not used in the 2D setting (if I'm not mistaken). Also, the number of slots predicted by the decoder seems an important hyperparameter. Furthermore, the number of vessel segments used to represent a vessel in the vessel autoencoder can be the subject of an ablation study.


### Minor remarks

1 - In the related works, the term RL is not introduced.

2 - In the introduction, the authors talks about image-to-geometry problems to motivate the interest of the presented method. I would have liked more examples related to the field of medical imaging and vessels analysis where the graph representation seems relevant.

---

> ### Author Response · Authors · 2025-03-08
> **Official Comment by Authors**
>
> - The reviewer requested more detail on the use of Fourier features in our approach and suggested an additional citation. In response, we explicitly clarified this at the beginning of the Methods section. We clearly described the use of sinusoidal Fourier features to effectively capture high-frequency geometric variations along vessel segments and included two additional relevant citations (Sitzmann et al., 2020 and Benbarka et al., 2022) to provide more context and justification for this choice. The Siren paper (Sitzmann et al.) describes how using periodic activation functions within neural networks are much better suited to learning functions with complex structure. The paper by Benbarka et al. "Seeing Implicit Neural Representations as Fourier Series" demonstrates that lifting coordinates to the fourier feature domain is analogous to using a hidden layer of a Siren network.
>
>
> - The reviewer requested clearly introducing and explaining the notation n and  \((x, y, z, r, t)\) for vessel segments at the start of the Methods section. We directly addressed this point by explicitly defining this notation at the beginning of the Methods section in the revised manuscript. Specifically, we clarified that \((x, y, z)\) represent spatial coordinates, \(r\) denotes the local vessel radius, and \(t \in [0,1]\) encodes the normalized position along each vessel segment. n describes the number of points down each input vessel. This clear introduction helps orient the reader and improves overall manuscript clarity.
>
>
> - The reviewer requested explicitly clarifying our definition of "topology" in the manuscript. We responded by explicitly defining "topology" at the start of the Methods section, clarifying that throughout our manuscript, the term refers specifically to discrete connectivity between nodes and edges (i.e., branching patterns), rather than algebraic topological invariants such as Betti numbers. This explicit clarification resolves any potential ambiguity around our usage of the term "topology."
>
>
> - The reviewer asked whether it is possible to apply our approach, specifically the Vessel Autoencoder, to a 2D dataset containing curvilinear centerlines. Although we did not explicitly discuss this scenario in the revised manuscript due to space constraints, we clarify here that this is indeed possible. Our current choice of the simplified 2D SSA dataset (straight edges only) was intentional to clearly demonstrate the method’s ability to handle purely topological connectivity information. However, creating a richer 2D synthetic dataset with curvilinear centerlines would naturally allow us to train a Vessel Autoencoder similarly to the 3D scenario, enabling the inclusion of continuous geometric embeddings ($z_v$) in the second-stage Vessel Tree Autoencoder. This represents an interesting direction for future investigation.
>
>
> - The reviewer mentioned relevant work in the space of graph neural networks. Although we acknowledge that some related approaches exist in molecular graph literature (such as the Junction Tree Variational Autoencoder for Molecular Graph Generation https://arxiv.org/abs/1802.04364), these methods are highly specialized for molecular structures. Adapting such methods to vessel tree geometry would require significant modifications, particularly due to fundamental differences in representation, geometry, and constraints specific to anatomical vessels. We agree with the reviewer that exploring such adaptations could be valuable but is beyond the scope of the current paper.
>
>
> - The reviewer suggested performing detailed ablation studies to better understand the contributions of individual model components. In the revised manuscript, we explicitly acknowledged this suggestion in the Results and Discussion section, stating that rigorous ablation studies and further analyses of model components are important next steps for future work. During development, we conducted preliminary ablations to arrive at the current architecture—for example, empirically determining that 32 slots in the decoder provided an effective balance between stability and computational cost. More systematic ablations will be valuable future investigations.

---

> ### Author Response · Authors · 2025-03-08
> **Official Comment by Authors**
>
> - The reviewer suggested explicitly defining the abbreviation "RL" (Reinforcement Learning) in the Related Work section. We have now clarified this in the revised manuscript by explicitly stating upon its first usage that "RL" refers to reinforcement learning, ensuring improved clarity for readers.
>
>
> - The reviewer suggested adding more examples from medical imaging and vessel analysis literature where graph representations have demonstrated relevance. Although we could not explicitly add this to the manuscript due to space constraints, we fully agree with the reviewer’s suggestion. Relevant examples for patient-specific blood flow simulations include: "Numerical Simulation of Blood Flow in an Anatomically-Accurate Cerebral Venous Tree" https://ieeexplore.ieee.org/abstract/document/6290398, "A One-Dimensional Hemodynamic Model of the Coronary Arterial Tree" https://www.frontiersin.org/journals/physiology/articles/10.3389/fphys.2019.00853/full

---

### Official Review · Reviewer_gDs5 · 2025-02-21

**Confidence:** 4
**Preliminary Rating:** 4
**Final Rating:** 4

**Summary:**

The authors introduce the vector representation of tree structures. The vector representation, especially for 3D tree structures, is efficient. Furthermore, the latent feature-based vector representation supports shape interpolation. Experiments on a 2D synthetic tree dataset and a 3D coronary artery dataset demonstrate the reconstruction fidelity.

**Strengths:**

The authors attempt to address the challenging topological shapes of tree-like structures, and the proposed method is well illustrated through the two-stage encoding using the VAE and recursive decoder. The details of the method are thoroughly discussed, and the experimental results demonstrate that topology-preserved results have been achieved.

**Weaknesses:**

My major concerns are as follows:
1.	Why are the vector embeddings powerful representations, and what is their clinical value for 3D coronary arteries?
2.	The practical use of vector representations is not discussed in sufficient detail.
3.	The comparison among GDVM-VAE, Conv-3D VAE, and VeTTA-VAE is not qualitatively described. It is unclear why the Dice performance shows such a difference, while the difference in mesh HD may not be significant.
Please see the detailed comments.

**Detailed Comments:**

The authors claim that vector embeddings are powerful representations. However, the experiments use mesh representations derived from cardiac CTA. The mesh representation is quite efficient compared to volumetric CTA. Additionally, the mesh representation can highlight patient-specific shape abnormalities, such as stenosis in the coronary arteries. Given the advantages of mesh representations, what benefits does the vector representation offer? How can the vector representation distinguish patient-specific differences for personalized medicine?

In practice, acquiring a patient's mesh immediately can be challenging. How can the proposed vector representation be used directly from the CTA images? This capability could be considered a truly powerful and compact representation.

Figure 2 only reports the results of the proposed methods; however, the results of the comparative methods, GDVM-VAE and Conv-3D-VAE, are not included. These methods show relatively low Dice scores compared to mesh-based measurements, and further discussion should be provided. Additionally, the comparison of the overlay between the ground truth centerline (yellow) and the reconstructed centerline is unclear. Is the author claiming that the reconstructed centerline is close to the ground truth, or is the claim that the reconstructed centerline is even better? Furthermore, since topology is broadly discussed in the paper, some topological measures, such as Betti errors, should be included.

**Justification Of The Final Rating:**

The authors have made sufficient improvements in the revision, particularly in clarifying the motivation and clinical value of the vector embedding, as well as providing a detailed comparison with other methods such as GDVM-VAE and Conv-3D-VAE. Although the discussion on personalized medicine in the context of vector embedding has not been quantitatively demonstrated, I acknowledge that it represents a valuable direction for future research.

**Justification Of The Preliminary Rating:**

The method is well illustrated to address the challenging topological shapes, and the paper is well-written. More discussion on the clinical and practical value of the vector representation should be considered.

**Questions To Address In The Rebuttal:**

As mentioned earlier, the explanation of why vector embeddings are powerful in this paper should be further discussed, and the comparison should be clarified.

---

> ### Author Response · Authors · 2025-03-08
> **Official Comment by Authors**
>
> - The reviewer asked why vector embeddings are powerful representations and requested clarification regarding their clinical value for 3D coronary artery analysis. In response, we expanded the Introduction of the revised manuscript, explicitly stating that vector embeddings provide compact representations ideal for direct neural network integration. We further highlighted their clinical significance, noting that their representational efficiency and scalability enable more accurate predictions of vessel geometry directly from clinical images. This has practical clinical relevance by potentially improving downstream tasks, such as hemodynamic assessments and diagnosis of coronary artery disease.
>
>
> - The reviewer asked about the specific benefits offered by our proposed vector representation compared to mesh representations. To clarify this, we explicitly addressed this point in the Results and Discussion section of our revised manuscript. We emphasized that, although mesh representations are efficient relative to volumetric CTA data, their complexity—typically consisting of thousands of nodes and edges—poses challenges for direct integration into downstream neural-network-based analyses. By contrast, our proposed vector representation provides a compact, fixed-length embedding of the entire vessel tree, greatly simplifying neural processing. Additionally, our method produces continuous geometric reconstructions, potentially capturing subtle anatomical details that would otherwise require extremely high-resolution meshes.
>
>
> - The reviewer asked how the proposed vector embeddings could distinguish patient-specific differences relevant to personalized medicine. Although we did not explicitly add this discussion to the manuscript due to space constraints, we clarify here that the accuracy of our reconstructions strongly indicates that the learned vector embeddings encode detailed geometric properties of individual patient vessel trees. As a result, clinically relevant, patient-specific differences derived from geometry (such as anatomical variations or pathology-related features) are likely represented in these vector embeddings. In practice, one could use simple linear classifiers or regressors directly on these embeddings to identify or quantify patient-specific characteristics relevant to personalized medicine. This represents a valuable direction for future investigation.
>
>
> - The reviewer asked how the proposed vector representations could be used directly from CTA images. In response, we clarified in the revised Introduction that one key motivation for using compact vector embeddings is precisely their suitability for direct integration with neural networks trained on clinical imaging modalities. In a more specific example, our proposed approach could serve as a "teacher" model trained on datasets containing explicit vessel geometry (such as VesselTrees), subsequently guiding an image-based model (e.g., convolutional or vision transformer architectures) to directly predict these embeddings from CTA images. The recursive decoder can then be used to reconstruct high-quality, patient-specific vessel geometries directly from clinical images. Thus, our approach represents a practical foundation for developing end-to-end image-to-geometry pipelines with clinical utility.
>
>
> - The reviewer requested a qualitative comparison between GDVM-VAE, Conv-3D-VAE, and our proposed VeTTA method. In response, we have explicitly addressed this in the revised manuscript by adding qualitative commentary in the Results and Discussion section. Specifically, we noted that the vessel reconstructions obtained with our VeTTA approach generally appeared more realistic, exhibiting smoother and more anatomically plausible vessel geometries compared to the convolutional baselines. The baseline methods frequently resulted in disconnected or topologically inaccurate reconstructions, highlighting clear qualitative advantages of our vector-based approach.
>
>
> - The reviewer asked why the Dice scores showed substantial differences, whereas differences in mesh Hausdorff distances (HD) appeared less pronounced. We addressed this explicitly in the Results and Discussion section of the revised manuscript, clarifying that Dice scores aggregate the overall volumetric overlap and are thus less sensitive to isolated errors. Conversely, Hausdorff distances are particularly sensitive to outliers, meaning occasional reconstruction errors disproportionately impact this metric. Hence, the Dice scores provide an overall comparison of reconstruction performance, while HD highlights the performance of outlier samples.

---

> ### Author Response · Authors · 2025-03-08
> **Official Comment by Authors**
>
> - The reviewer asked for clarification regarding the purpose of the centerline overlay visualization shown in Figure 2. To address this explicitly, we have updated the figure caption in the revised manuscript, clearly stating that the overlay visualization (right) is intended to illustrate the close anatomical alignment between the reconstructed centerline (red) and the ground truth centerline (yellow). This clarification makes it explicit that our goal was to demonstrate accurate reconstruction, not superiority over the ground truth.
>
>
> - The reviewer suggested including topological measures such as Betti numbers in our evaluation. We appreciate this comment but would like to clarify our terminology explicitly. Throughout this manuscript, we use the term "topology" strictly to refer to discrete connectivity between nodes and edges (e.g., branching patterns), rather than algebraic topological invariants such as Betti numbers, which characterize holes or voids in continuous topological spaces. We have added a clear statement at the beginning of the Methods section to explicitly define our usage of "topology" and prevent future confusion.

---

### Official Review · Reviewer_MVrx · 2025-02-22

**Confidence:** 3
**Preliminary Rating:** 4
**Recommendation:** Oral
**Final Rating:** 5

**Summary:**

The authors introduce a framework to learn vector representations of vessel trees. It relies on 2 transformer-based autoencoders. The first one deals with geometry and learns representations of a vessel branch, represented as a generalized cylinder. The second one deals with topology and learns representations for trees. Autoencoder and Variational Autoencoder variants are studied. Experiments are reported on both a 2D synthetic dataset, compared to conventional convolutional 2D encoders, and a 3D dataset of coronary artery shapes extracted from CTA patient data, compared to a 3D CNN and 2 other convolution architectures dedicated to vessel segmentation. Various ensemble, mesh, and centerline distances demonstrate the improved performance of the proposed archiecture.

**Strengths:**

The proposed framework is able to handle both geometry and topology in a compelling way. The paper is rather well structured and figures are informative (in particular figure 1). Experiments are performed on both synthetic and actual patient data, on large databases, with clear results compared to alternatives approaches.

**Weaknesses:**

- My first concern is the length of the appendix (14 pages), together with several aspects of the paper that lack precision and details. Hence the appendix is required to be read in order for the paper to be understood.

- Some aspects of the experiments are unclear, in particular on 3D data. Details are missing concerning the VesselTrees database.

- Comparing only to convolutional segmentation networks, makes it difficult to discriminate the influence of post-processing operations that are necessary e.g. to extract the centerline.

**Detailed Comments:**

My first concern is about the length of the appendix. This makes it difficult to believe the authors will be able to provide the audience with a synthetic presentation of their work. Reading the 14 pages appendix is mandatory to understand several aspects of the work. The authors might consider refocusing their work and reformat the proposed paper for a journal paper (completed with more experiments).

The authors should make it clear from the beginning that
- the proposed framework can be applied to both 2D and 3D problems. For example, a obscure mention of "in the 3D setting" is made on page 4, when all previous exposure, including Figure 1, was about 3D data.
- two variants are proposed, one with regular autoencoders and a second one with variational autoencoders. This is not mentioned before page 4 and equation 3.

The general impression is that too many things are forcibly squeezed into the tiny space of the paper. Details should be provided in the text about
- the m function (equation 1)
- the topological attributes mentioned at the top of page 4
- the skip-vessel flag (page 4)
- the recursive reconstruction loss (pages 4 and 5)

The fact that the VesselTrees dataset provides the centerlines for the vessel is an asset for the proposed method whereas other CNN based models a priori only exploit the voxelized meshes. Could the authors comment on this potential source of unfair comparison?

During tree reconstruction, a vessel can be continued with 0, 1 or 2 vessels. The cases of 0 vessel is clear: this stops the reconstruction.

The case of 2 vessels is also clear, but it raises the question of the anatomical quality of the reconstructed tree. The 2D SSA datasets was built to satisfy anatomical properties, in particular Murray's law. The 3D dataset contains patient data and thereafter also satisfies such properties. What about the vessel tree reconstructed by the proposed framework? Are the induced representations able to encode such properties?

The case of 1 vessel is not really addressed in the paper, though it can be questioned. Was it necessary because the representations are unable to encode beyond a certain amount of curvature and torsion variations along the vessel (limited tortuosity)? How good is the continuity between successive vessels in that case?

Generally speaking questions arise concerning the complexity of the vessel tree that can be reconstructed in 3D. Beside curvature, torsion should be addressed. Also, complex trees with many levels of bifurcations are shown in 2D (e.g. figure 3). What about 3D trees? Is there a limit to the realism of the tree beyond a couple of bifurcations? Could this be applied to liver or brain vasculature (knowing that the tree hypothesis is limited in those cases)?

The demonstration of the model to provide realistic shapes when interpolating in the latent space is definitely interesting. Is it necessary to devote part of the limited writing space to demonstrate it, though?

**Justification Of The Final Rating:**

I appreciate the authors' involvement and efforts in improving the clarity of their manuscript and making it more self-contained. I also enjoyed reading their insights on the capabilities and limitations of their model. I do consider the proposed model is original and particularly suited to vascular modeling, and makes even more general sense to represent both geometry and topology. As a consequence, I raise my rating to Strong Accept and think the paper has the potential to be of interest to a large audience should therefore be considered for an oral presentation.
But, I still want to emphasize that, in my opinion, authors in general, should restrain from appending long sections to their main manuscript. First, papers should be self contained, and MIDL reviewers are indeed not required to read appendices. Second, this is an equity issue towards other authors that strive to adhere to the page limitations, though I reckon that the vast majority of these authors have more than enough material that they would love to share and discuss. I do believe that the proposed work can be quite straightforwardly extended into a journal publication.

**Justification Of The Preliminary Rating:**

This is a very good work, with many contributions. However, I feel the will to push all contributions in a single 8 page paper is detrimental to its readability and the appreciation of the proposed approach. My advice would be to adapt the publication strategy here and focus on what the authors consider to be the most compelling aspect of their work, and take the time to demonstrate its interest with more in depth experiments and discussion.

**Questions To Address In The Rebuttal:**

- Improve readability, with less dependence on the appendix, e.g. by focusing on the most important aspect of the work. Reduce the length of the appendix. Provide more details about unclear part of the text (see detailed comments)
- Comment or provide experimental results, on the anatomical fidelity of the reconstructed trees in 3D (Murray's law, continuity, tortuosity)

**Special Issue:**

Yes

---

> ### Author Response · Authors · 2025-03-08
> **Official Comment by Authors**
>
> - We appreciate the reviewer’s feedback regarding the length and readability of our paper and appendix. To address this, we leveraged the additional page allowed in the final submission (increased from 8 to 9 pages) to integrate key explanations previously located in the appendix into the main text. Specifically, we clarified the notation for vessel representations and explicitly introduced Fourier features at the start of the Methods section. We also clearly distinguished the 2D and 3D experimental settings to avoid ambiguity, and provided an expanded description of the VesselTrees dataset in the Experimental Setup section. While some redundancy between the main text and appendix remains, these revisions improve readability and ensure key details are accessible directly within the manuscript.
>
>
> - The reviewer asked for additional clarity regarding the VesselTrees dataset. In response, we expanded the description within the Experimental Setup section. The description explicitly clarifies the dataset’s composition (full coronary trees versus subsampled sub-trees), dataset splits (3996 training, 754 testing, and 250 validation samples), and structure. The handling of short vessel segments with a skip-vessel flag is additionally mentioned in the methods section.
>
>
> - The reviewer noted ambiguity regarding our use of the term "in the 3D setting." To address this clearly, we explicitly defined the differences between the 2D and 3D experimental settings at the start of the Methods section. Specifically, we clarified that the 3D setting (exemplified by the VesselTrees dataset) encodes full curvilinear centerlines and explicit radius information through a dedicated Vessel Autoencoder, resulting in detailed continuous geometric embeddings (z_v). In contrast, the 2D setting was clarified as only involving directed edges without these additional geometric details.
>
>
> - The reviewer suggested clarifying earlier in the manuscript that two variants (standard and variational autoencoders) are evaluated in both the 2D and 3D settings. We addressed this by explicitly adding introductory paragraphs at the start of the Methods section, clearly stating that both Autoencoder (AE) and Variational Autoencoder (VAE) variants are proposed and evaluated across both experimental settings. This clarification ensures readers are aware of both model configurations upfront.
>
>
> - The reviewer requested further detail on the function $m(t)$ appearing in Equation (1). We have explicitly clarified this in the revised manuscript. Specifically, during training, we set $m(t) = 1$ to ensure stable optimization, whereas during evaluation, we define $m(t)$ as:
> \[
> m(t) = 0.5^{-0.2}\cdot t^{0.1}(1 - t)^{0.1},
> \]
> ensuring exact matching at vessel segment endpoints ($m(0)=m(1)=0$). This facilitates precise endpoint alignment and smooth curvature reconstruction.
>
>
> - In response to the reviewer’s request for clarity regarding the "topological attributes," we explicitly detailed this in the method section. Specifically, we clarified that topology explicitly refers to discrete connectivity between nodes and edges. This clear definition removes ambiguity about our use of the term "topology," aligning precisely with the previously provided information in the appendix.
>
>
> - The reviewer asked for additional clarity regarding the skip-vessel flag. In response, we explicitly clarified this in the revised manuscript. We stated that the skip-vessel flag is specifically introduced to handle short vessel segments (<2.5mm). Edges marked with the skip-vessel flag have their vessel embeddings excluded from the loss computation during training. During inference, these short segments are reconstructed by linearly interpolating between their endpoint properties without additional curvature refinement, ensuring accurate handling of these cases.
>
>
> - The reviewer requested additional details regarding the recursive reconstruction loss used in the Vessel Tree Autoencoder. Although we were unable to integrate the full description directly into the manuscript due to space limitations, we provide a brief summary here. Specifically, the recursive reconstruction loss involves constructing a cost matrix by computing pairwise distances between predicted slot vectors and the lifted representations of ground-truth target nodes. A custom two-way matching algorithm (top-k matching) is applied at each decoding step, ensuring each predicted slot matches exactly one ground-truth node, and each target node matches at least k=3 predicted slots. This recursive strategy guarantees accurate and topologically consistent reconstructions from a single global embedding.

---

> ### Author Response · Authors · 2025-03-08
> **Official Comment by Authors**
>
> - The reviewer raised an important point regarding the potential source of unfair comparison arising from the difference in input data representation between our proposed model (explicit centerline geometry) and convolutional baseline methods (3D voxelized masks). We explicitly acknowledged this point in the Results and Discussion section, clearly stating that our Vessel Tree Autoencoder directly leverages centerline geometry, whereas convolutional baselines rely on voxel-based inputs and therefore require additional skeletonization post-processing steps. We emphasize this intrinsic difference indeed provides a representational advantage to our method, particularly benefiting metrics related to centerline reconstruction accuracy.
>
>
> - The reviewer suggested discussing anatomical properties, particularly adherence to Murray’s law, of the reconstructed vessel trees. Although we could not explicitly integrate this discussion directly into the revised manuscript due to space constraints, we would like to address this here. Qualitatively, we observe that the reconstructed vessels demonstrate realistic tapering and branching patterns consistent with physiological expectations such as Murray’s law. We agree with the reviewer that future quantitative analysis to explicitly measure adherence to Murray’s law at bifurcations would be valuable.
>
>
> - The reviewer raised an insightful point regarding the applicability of our method to single long vessels composed of consecutive segments, each with a single child. Our proposed method can indeed handle this scenario. However, for clarity and focus, we specifically restricted our experiments to samples containing at least one bifurcation, clearly framing the work within the context of tree-structured vessel geometries. In preliminary experiments, we observed that encoding single vessel segments exceeding 40mm with the Vessel Autoencoder led to reduced reconstruction accuracy. We hypothesize that, beyond a certain level of complexity and vessel length, it becomes challenging for the Vessel Autoencoder to represent all necessary geometric information accurately within a single fixed-size embedding (dimension 64). This motivated our design decision to limit individual segments to 40mm and apply loss function reweighting (as detailed in Appendix A.4) to enhance performance on longer or more tortuous vessels. Importantly, our two-stage Vessel Tree Autoencoder approach mitigates these limitations by hierarchically pooling multiple vessel segment embeddings into a substantially larger embedding vector (dimension 8192). The recursive decoding process inherently ensures continuity between consecutive vessel segments, thus maintaining accurate and realistic reconstructions even for extended vessels divided into multiple segments.
> While demonstrating this explicitly would require creating a new, carefully curated dataset and re-running extensive training and evaluation procedures (which is beyond the scope of this rebuttal), we agree it is an interesting avenue for future exploration.
>
>
> - Regarding the reviewer's query about vessel torsion, we clarify that our current representation encodes vessels as generalized cylinders parameterized by coordinates (x,y,z) and radius r at each point along their centerline. Our representation does not explicitly encode torsion because this would require an orthonormal frame defined at each point along the vessel, which is unnecessary for our chosen surface reconstruction method (Poisson reconstruction), as surface points are fully determined by the centerline coordinates and radius information alone. In principle, our approach could be adapted to explicitly include torsion by adding two orthogonal basis vectors defining a local frame around each centerline point. Such an adaptation would enable quantification of vessel torsion as rotations of this frame along the vessel path. However, this is beyond the scope of this work and is therefore suggested for potential future investigation.

---

> ### Author Response · Authors · 2025-03-08
> **Official Comment by Authors**
>
> - The reviewer highlighted the question of scalability, particularly concerning whether our method could generalize to full coronary trees, which typically have significantly more bifurcations than those evaluated in the VesselTrees dataset. We agree that, in principle, the approach proposed in this paper is applicable to larger, more complex coronary artery trees. Indeed, our 2D experiments demonstrate that our method effectively encodes complex topological structures into fixed-length vectors. However, accurately encoding full coronary trees would likely require further method enhancements, including larger transformer models, increased latent dimensions, and larger training datasets. We anticipate that our current model, benefiting from the favorable scaling properties inherent in transformer architectures, could be adapted to handle larger-scale data given additional computational resources. At our current computational scale, however, we expect the proposed implementation would struggle to represent highly complex vessel geometries significantly more intricate than those considered in our current experiments. Beyond the tree-structured vasculature, the method also holds promise for generalized graph or set-structured geometries, though implementing such generalizations would require further adaptation beyond the scope of this work. Our goal in this paper has specifically been to clearly demonstrate the potential of our approach within the constraints of strictly tree-structured vasculature.

---

### Author Rebuttal · Authors · 2025-03-08

**Rebuttal:**

We would like to thank the reviewers for their constructive and valuable feedback, and we greatly appreciate the support for our work. We have done our best to incorporate as many suggestions as possible within the time frame available for revisions. We have uploaded a revised version with changes clearly highlighted in colored text for ease of review. Detailed responses addressing each specific comment can be found below.


We acknowledge reviewers' concerns regarding the length of the appendix. In preparing the appendix, our goal was to provide comprehensive information that would enhance transparency, facilitate reproducibility, and serve as a valuable resource for the broader research community. While we understand that the appendix is extensive, we believe these details are essential and there would not be a benefit to omitting them from the final version. We therefore respectfully suggest retaining this detailed supplementary material to support future researchers interested in reproducing or extending our results.

**Supporting Material:**

/attachment/99353f01ec96b0df9661f8888ccba0534ba2dfa9.pdf

---

### Meta-Review · Area_Chair_8fiM · 2025-03-21

**Recommendation:** Accept (Poster)
**Confidence:** 4

**Metareview:**

This paper proposes a transformer-based framework for learning vector representations of vascular trees, combining geometric and topological information. It employs a two-stage autoencoder, with one stage focusing on local geometry and the other on global structure. The recursive design ensures tree structure preservation. The method shows strong performance, outperforming baseline convolutional models.

Strengths:
- Interesting concept and novel approach.
- Robust performance with topology preservation.

Weaknesses:
- Some technical details lack clarity, and the appendix is excessively long.
- Limited comparison with other graph-based methods.

Overall, the paper clearly presents an interesting method, but it also has some shortcomings. Besides the issues pointed out by the reviewers, I would like to mention two additional concerns:

1. The proposed method heavily relies on preprocessing and is tailored to the specific application in the paper. For more complex structures, such as brain vasculature, the method may face significant challenges and would likely require substantial adjustments. While the recursive approach is theoretically elegant, it is not very friendly to deep learning methods, especially when there is a large scale variation.

2. The comparison methods used are somewhat naive. Apart from the lack of comparisons with other graph-based methods, the paper does not even consider sparse methods, such as sparse convolutions. Since the proposed method inherently has sparsity, it seems unfair not to include comparisons with sparse convolutional approaches.

A comment on terminology:
The use of the term "vector" immediately made me think of vector graphics. I initially expected a paper that would finally introduce vector graphics into medical applications. However, the method here essentially represents a neural or latent representation rather than a true vector graphic. Considering that "vector" already has an established meaning in graphics, this choice of terminology seems somewhat misleading.

While these shortcomings do not prevent the paper from being a strong contribution and deserving acceptance, I do not find it sufficient for an oral presentation.